# Protocol to Assess the Knowledge, Attitude, and Practices of Midwives in the Implementation of Maternal Healthcare Guidelines in a Selected District, Limpopo Province, South Africa

**DOI:** 10.3390/nursrep15100368

**Published:** 2025-10-15

**Authors:** Mantsha Sarah Maeko, Thifhelimbilu Irene Ramavhoya, Tebogo Maria Mothiba, Mamoeng Nancy Kgatla, Nonkululeko Peaceful Ntshayintshayi

**Affiliations:** 1Department of Nursing Science, School of Health Sciences, University of Limpopo, Polokwane 0727, South Africa; irene.ramavhoya@ul.ac.za (T.I.R.); nancy.kgatla@ul.ac.za (M.N.K.); peaceful.ntshayintshayi@ul.ac.za (N.P.N.); 2Office of the DVC: Research, Innovation and Partnership, University of Limpopo, Polokwane 0727, South Africa; tebogo.mothiba@ul.ac.za

**Keywords:** attitudes, antenatal care, knowledge, implementation, midwives, practice, primary healthcare facilities, maternal healthcare guidelines

## Abstract

Maternal healthcare guidelines (MHCGs) are documents with detailed information on how to manage maternal and perinatal conditions step by step and accordingly to reduce preventable maternal and perinatal deaths. The National Maternity Guidelines Committee in the Department of Health, guided by the WHO, developed the (MHCGs) to ensure that every primary healthcare facility (PHC) has one or two midwives at their disposal to use, but its implementation remains a huge problem in South Africa. The purpose of this study is to evaluate the knowledge, practices, and attitudes of midwives during the implementation of MHCGs in a selected district, Limpopo province, South Africa. A quantitative methodology will be employed; cross-sectional descriptive research design will be used. The population of this research study will be midwives working in the PHCs of a selected district, Limpopo province, South Africa. The convenient sampling approach will be used; whereby self-developed semi-structured questionnaires will be used as a data collection instrument. The collected data will be analyzed using the Statistical Package for Social Sciences (SPSS), version 29, with the help of a statistician. The results will be available after data collection. The conclusion and recommendations will be based on the findings of the study.

## 1. Introduction

The global public health system faces the constant challenge of maternal death from pregnancy and childbirth complications. In 2020, approximately 287,000 women died during childbirth worldwide, with 800 of those deaths occurring every day from preventable pregnancy and causes related to childbirth [1]. Globally, the death of a pregnant woman is reported every eight minutes, either due to difficulties during pregnancy, labor, or the postpartum period [2]. Women in Sub-Saharan Africa face a 47-fold higher mortality rate than women in the United States, which puts them at risk of dying [3]. Additionally, the majority of maternal deaths globally occur in Sub-Saharan Africa, and the majority of them are caused by limited access to antenatal care [4].

The findings of the study by Mahada, Tshitangano, and Mudau (2023) [5] revealed that a high maternal mortality rate has been associated with many systematic factors in low- and middle-income countries, such as poor facilities and a lack of personnel. In low- and middle-income nations, pregnancy-related causes claim the lives of thousands of women each year, attributed to poor management and the inability to follow protocols and maternal health guidelines (MHCGs), regardless of their availability [5]. Ramavhoya, Maputle, Ramathuba, Lebese, and Netshikweta (2020) [6] also supported the above-mentioned findings by revealing that although MHCGs are available to midwives, a shortage of human resources had a negative effect on their implementation. Ward (2014) [7] discovered that midwives were reluctant to apply MHCGs when managing pregnant women; therefore, they managed the women based on previously learned knowledge, which is the knowledge learned during their basic midwifery training, regardless of the availability of MHCGs. The World Health Organization (WHO), together with other international organizations, including the United Nations Population Fund (UNFPA), has created a series of evidence-based guidelines for the management of maternal diseases such as PMTCT, WHO recommendations for the prevention and treatment of PPH, and WHO recommendations for the prevention and treatment of preeclampsia [8]. MHCGs are commonly adopted, altered, or contextualized in different nations. In Kenya, for example, the Ministry of Health, in collaboration with its partners, leads the effort to contextualize WHO MHCGs for the Kenyan context. The Kenyan MHCGs addressed this by focusing on how the recommendations will be communicated [9]. However, it has not addressed the effectiveness of the tactics indicated in the distribution and implementation of MHCGs.

Despite the existence of several measures to improve the quality of care, the South African maternal mortality rate is still high. Between 2020 and 2021, the maternal mortality rate increased by 30%, with Limpopo having the fourth highest rate among nine provinces [5,10]. According to the Saving Mothers report (2023) [11], the Capricorn district falls under the top five districts with respect to maternal deaths, with a total number of 48 maternal deaths [10]. Most maternal deaths that could be avoided and prevented were attributed to poor management skills by healthcare professionals, such as poor evaluation, treatment, and admission delays [12,13]. Most of the maternal mortality rate was reported during the postnatal period; however, according to the Saving Mothers report (2018) [14], little is known about the benefits of using South African MHCGs in terms of postnatal care by midwives.

Although midwives are important workers in the healthcare system, providing postnatal care to both women and babies, various hurdles influence how well-informed they are about MHCGs [14,15,16]. The study by Ntuli, Mogale, Hyera, and Naidoo (2017) [17] revealed that a review of all maternal mortalities occurred at Pietersburg Hospital over a period of five years from 2011 to 2015, indicating a total of 14,685 live births and 232 deaths of mothers. To end the international problem of preventable deaths in low- and middle-income countries, the use of MHCGs and cost-efficient healthcare is urgently needed [8]. As such, this study will be conducted to determine the knowledge, attitudes, and practices of midwives in the implementation of MHCGs. The findings will address the gap by educating and encouraging the use of MHCGs during practice and reducing the maternal mortality rate in a selected district, Limpopo province, South Africa.

### Conceptual Framework

This study is guided by a conceptual framework that examines the relationship between midwives’ knowledge, attitudes, and practices (KAP) and the quality of midwifery care provided within primary healthcare settings, and how these influence the wellbeing of pregnant women and fetuses. The framework integrates core concepts central to maternal health service delivery: midwives, midwifery care, primary healthcare, and the wellbeing of pregnant women and fetuses, anchored by the implementation of maternal healthcare guidelines (Figure 1).

Primary Healthcare as the Delivery Context

Primary healthcare serves as the foundational context for the provision of maternal health services. It is characterized by its emphasis on accessibility, continuity, community orientation, and comprehensiveness [18]. Within this setting, located in the selected district, midwives play a frontline role in delivering maternal healthcare, including antenatal, intrapartum, and postnatal services, using guidelines, protocols, and other resources as guidance.

2.Midwives as Key Providers (Agents)

Midwives who are working in the PHC facilities of the selected district are central to maternal healthcare and act as primary implementers of national and international maternal healthcare guidelines. Their professional competencies, clinical judgments, and interpersonal interactions significantly influence maternal and fetal health outcomes [19].

3.Knowledge, Attitudes, and Practices (KAP) of Midwives

Rogers’ knowledge, attitudes, and practices (KAP) model, proposed in the late 1950s and early 1960s, is employed to understand the conceptual framework on how midwives engage with maternal healthcare guidelines. Each element will be applied as follows:Knowledge refers to the midwife’s understanding and awareness of evidence-based maternal healthcare guidelines, including clinical protocols and best practices.Attitudes involve personal beliefs, values, and the level of acceptance or resistance towards the application of the guidelines in daily practice.Practices denote the actual implementation of maternal healthcare guidelines during patient care, reflecting the extent to which midwives apply their knowledge and attitudes in clinical settings.

The KAP elements are interrelated and collectively influence the effectiveness and consistency of midwifery care.

4.Midwifery Care (Process)

Midwifery care encompasses a holistic and patient-centered approach to the health and wellbeing of pregnant women attending maternal health care services at the selected district of Limpopo Province. It includes preventative, supportive, and curative interventions during pregnancy, childbirth, and the postnatal period. The quality of care delivered is contingent upon the extent to which midwives adhere to established maternal healthcare guidelines, which in turn is shaped by their knowledge, attitudes, and practices.

5.Wellbeing of Pregnant Women and Fetuses (Outcome)

The ultimate outcome of interest in this framework is the wellbeing of both pregnant women and their fetuses. This includes physical health (e.g., absence of maternal or neonatal complications), psychological wellbeing, and satisfaction with the care received [20]. High-quality, guideline-based midwifery care, informed by midwives’ adequate knowledge and positive attitudes, is expected to improve maternal and fetal health outcomes of pregnant women who are attending maternal health care services in a selected district in Limpopo Province.

6.Assumptions about the Conceptual Framework

This conceptual framework posits the following causal pathways:Midwives’ knowledge influences their attitudes toward maternal healthcare guidelines.Positive attitudes enhance the likelihood of appropriate clinical practices.Effective implementation of maternal healthcare guidelines through midwifery care leads to improved maternal and fetal wellbeing.The primary healthcare setting facilitates this process by providing a structured environment for continuity and accessibility of care (Figure 1) below.

Figure 1 below represents the Conceptual Framework, which illustrates the Influence of Midwives’ Knowledge, Attitudes, and Practices on the Implementation of Maternal Healthcare Guidelines and subsequent wellbeing outcomes.

7.Relevance to the Study

This conceptual framework provides a logical and empirical foundation for examining the role of midwives in improving maternal health outcomes. It highlights the critical need to assess and strengthen midwives’ knowledge, attitudes, and practices as a pathway to improve the implementation of maternal healthcare guidelines. The framework is particularly relevant in low- and middle-income settings where healthcare systems rely heavily on midwives in primary healthcare facilities.

## 2. Materials and Methods

### 2.1. Objectives

Assess the level of knowledge among midwives about MHCGs in a selected district in Limpopo Province, South Africa.To evaluate the attitudes of midwives towards the implementation of MHCGs in a selected district in Limpopo province, South Africa.Investigate the practices of midwives in relation to adherence to and application of MHCGs in the selected district of Limpopo Province, South Africa.Determine the association between sociodemographic data of midwives with knowledge, attitudes, and practices in the implementation of maternal healthcare guidelines in a selected district in Limpopo province, South Africa.

### 2.2. Research Methodology and Design

Quantitative research is the concept of quantity or extent. It relates to something that can be measured. It uses statistical, mathematical, or computational methods to analyze observable phenomena [21]. Quantitative research will be used in this study to determine the knowledge, attitudes, and practices of midwives in the implementation of MHCGs in the Capricorn district, Limpopo province, and frequencies and percentages will then be used in the table and figures to represent the data collected.

Research approaches are strategies and protocols that cover everything from general hypotheses to specific techniques for collecting, analyzing, and interpreting data [22].

The nonexperimental approach will be employed in this study. The researcher will gather information from a specific group, made up of midwives who work in PHC facilities that fall under a selected district, Limpopo province, South Africa. In this study, a descriptive cross-sectional design will be used. The researcher will use this design at one point in time to collect information on the knowledge, attitudes, and practices of midwives in the implementation of MHCGs in a selected district, Limpopo province, South Africa.

### 2.3. Setting and Population of the Study

The study will be conducted in one of the five municipal districts of Limpopo province, which consists of six hospitals and 96 PHC facilities. PHCs that fall under one district municipality will be selected to be part of the study, and they are 42 in total. The research will only focus on the 42 PHCs since they all offer maternal healthcare services and receive maternal healthcare guidelines. The facilities offer comprehensive health care services, including maternal health care services, although some of them focus on antenatal and postnatal care and only perform deliveries in emergencies and operate only during the day. Each facility consists, for the most part, of four to six midwives, along with the rest of the professional nurses, who are not midwifery trained but just general nurses. Each month, some facilities see plus or minus 30 pregnant women for antenatal care; and for deliveries, pregnant women are referred to the hospital. The population will consist of midwives who work in PHC facilities that fall under the selected district—Limpopo province, South Africa. The selected district has 42 PHC facilities that feed the provincial and academic hospitals, each consisting of four to six midwives, which makes the total population of this study 252 participants.

### 2.4. Sampling

Sampling is a method for choosing participants from the target population based on accessibility [23].

Purposive sampling will be used in this study because the researcher will choose the PHC facilities that fall under a selected district in Limpopo province, South Africa, since all these facilities directly feed the provincial and academic hospitals of that district; these hospitals had about 3–5 cases of maternal deaths per month in 2024, and all pregnancy-related problems are seen first at PHC facilities. Convenient sampling will be used in the selection of participants because the study requires many participants who are widely sparse and cannot be gathered in the same place at the same time due to the work they do. Therefore, the researcher will sample participants who are on duty at that time and those who are willing to participate. This will continue until the necessary number of participants is chosen. The sample size will be determined by making use of G Power software 3.1.97 with the help of a statistician.

### 2.5. Inclusion Criteria

All midwives who work in PHC facilities that fall under the selected district, Limpopo province, South Africa, with six months of experience, will be included in the study, since they have sufficient exposure to MHCGs and all offer maternal health services.

### 2.6. Exclusion Criteria

All midwives who are off duty during the data collection period and those who refuse to participate will be excluded from the study.

### 2.7. Pilot Study

By administering self-structured questionnaires to 15 midwives, which is 10% of the sampled midwives, to be chosen from PHC facilities in a selected district of the province of Limpopo, South Africa, the researcher will carry out the pilot study at facilities not chosen for the study; these facilities will fall under Molemole municipality, in Capricorn district—namely, Nthabiseng clinic, Eisleben clinic, Ramokgopa clinic, Botlokwa gateway, and Matoks clinic. Participants in the pilot study will not be involved in the main study and the results of their participation will not be incorporated into the main study. In this way, the researcher will be able to pretest the study and identify any problems with the research tools.

### 2.8. Data Collection

Data collection is the process of gathering information to learn more about the research problem [24]. The researcher will use self-administered questionnaires, which are short and straightforward. The questionnaires will be developed based on the literature collected and will be divided into three sections, whereby Appendix A will consist of demographic information from the participant, Appendix A will cover questions related to the knowledge of midwives about the implementation of MHCGs, and Appendix A will cover the attitudes and practices of midwives in the implementation of MHCGs. The help of a statistician and supervisors will be required to design the questionnaire to ensure its validity. Cronbach’s alpha will also be used to determine the reliability of the questionnaires. The researcher will recruit participants by first calling the operational managers of the sampled facilities and requesting that they inform their staff that the researcher will come and collect data on a day that is convenient for them and when they are not too busy. To collect the data, the researcher will visit each sampled participant at their place of employment and during their free time or on days that are not very busy, such as weekends, so that the day-to-day functioning of the facility is not disturbed. The researcher will explain the purpose of the study to the participants; consent forms will be explained thoroughly and signed before answering questionnaires. The questionnaire will be created in English and since the participants can read and write in English, the researcher will not translate the questions. To measure the knowledge, attitudes, and practices of midwives, the Likert scale, yes or no, and semi-structured questions will be used to design the questionnaires. The researcher will help clarify any questions that the participants do not understand. To ensure the collection of high-quality data, the researcher will ensure privacy by remaining in the other room for participants who feel uncomfortable with the researcher being there while they answer the questions. On the questionnaire, participants will need to select their response by ticking the boxes for yes or no, true or false, agree or disagree. It will take 20–25 min for participants to complete the questionnaire. The researcher estimates that six weeks is enough for the data collection.

### 2.9. Data Analysis

Data analysis is essentially the process of transforming the acquired data into a relevant descriptive approach, since it uses metrics like frequency distributions and information [23]. The questionnaire is divided into three sections: Appendix A; Appendix A; and Appendix A. Data will be coded and entered into SPSS (Statistical Package for the Social Sciences) version 29 for analysis. Descriptive statistics including frequencies, percentages, means, and standard deviations will be used to summarize the demographic characteristics and responses to the closed-ended questions. For Appendix A. Descriptive statistics will be used to present the distribution of respondents by age, race, gender, nationality, years of experience, and prior exposure to the South African maternal healthcare guidelines. Frequencies and percentages will be presented in tables and charts. Where applicable, cross-tabulations will be performed to examine potential relationships between demographic variables and other study variables. Appendix A Responses in this Appendix A will be analyzed using both frequency distributions and summary statistics for ordinal responses (Likert scale items). Binary (Yes/No) questions such as exposure to the guidelines, availability of the latest version, and training received will be summarized using frequencies and percentages. Likert-scale items (e.g., frequency of consulting guidelines, ease of use) will be analyzed using mean ranks, medians, and mode values. To test for associations between demographic factors (such as years of experience or training received) and knowledge levels, Chi-square tests or Spearman’s correlation will be used, depending on the level of measurement. Appendix A—this section contains several Likert-scale statements reflecting attitudes and practices related to the use of maternal healthcare guidelines. Responses will be analyzed using descriptive statistics, and the internal consistency of the scale will be evaluated using Cronbach’s alpha. Correlational analysis will be performed to determine whether there is a relationship between midwives’ attitudes and their reported use of the guidelines in clinical practice. The information will be kept on password-protected computers and USB drives.

### 2.10. Validity and Reliability

Validity is determined by what an instrument measures and how successfully it does so, and reliability refers to the accuracy of the data collected and the extent to which any measurement instrument could identify the error [25]. Face, content, and construct validity will be included in this study.

#### 2.10.1. Face Validity

To ensure face validity in this study, experts in maternal health care services will receive a questionnaire to measure the face validity instrument.

#### 2.10.2. Validity of the Content

The statistician, together with the supervisors, will receive the questionnaire to check if the questionnaire is appropriate and if the instrument is accurate. The researcher will use the gathered literature and the theoretical framework to develop the questionnaire. A rating scale will be provided to measure the accuracy of the instrument.

#### 2.10.3. Construct Validity

To ensure construct validity, the researcher will precisely define concepts based on the theoretical framework and literature. A pilot study will be conducted to test the precision and relevance of the measurement tool.

#### 2.10.4. Reliability

Reliability refers to the consistency or stability of a measurement tool. When used consistently, a reliable device produces consistent results [26]. In this study, the researcher will use the same method and instructions throughout the study to ensure reliability. The researcher will also pilot test the questionnaires to check for consistency. The Cronbach Alpha will be used to determine the reliability of the questionnaires. The scores range from 0 to 1. The pretest results of the questionnaires will be used to test the scores, whereby a score of less than 0.69 will be unacceptable and a score of greater than 0.69 will be acceptable.

### 2.11. Bias

Bias is characterized as an unjustifiable direct causal impact [27]. Systemic bias and random bias are the two categories of bias. Pure chance, human volatility, and inherent differences in measurement device precision can all lead to random bias. Systemic bias arises in the design of the study, data collection, statistical analysis, interpretation of the results, and the publication process [28]. In this study, systemic bias will be prevented by making sure that all sampled participants work in maternity areas. The researcher will not help the participants answer the questionnaires and will also not share their experiences and expectations with the participants to ensure that no systemic bias occurs.

## 3. Ethical Consideration

A set of moral guidelines that direct the design, execution, assessment, and reporting of any research endeavor is known as ethics in research. It gives the researcher principles, standards, and norms for appropriate behavior toward study participants, co-researchers, research assistants, fieldworkers, the institution, and sponsors [29]. The researcher will obtain ethical clearance from the University of Limpopo Turfloop Research Ethics Committee (TREC).

### 3.1. Permission

The researcher will seek permission from the Limpopo province Department of Health, the District Executive Officer of the district’s PHC facilities, the assistance and operational managers of the sampled facilities, and the participants, who are professional midwives.

### 3.2. Informed Consent

Informed consent is a fundamental idea in modern, autonomy-based research and helps participants and researchers make decisions together [30]. Before gathering data, the researcher will obtain the consent of the participants so that each person can decide whether to participate in the study after being fully informed of it and taking the necessary precautions to ensure that they fully understand the procedures to be followed and the risks and rewards that will be involved. To ensure that consent is acquired, participants will receive a form to complete before participating in this study. The researcher will explain to participants that they are free to terminate the study by informing them that they can opt out at any point during the project, including while answering questionnaires, and that they will not be penalized or hurt at any point during the interaction, including after signing the consent form.

### 3.3. Confidentiality

The researcher will guarantee confidentiality by making sure that the data that participants contribute are not disclosed to third parties outside of the research project. The data obtained will be stored on a computer in an encrypted file that can only be accessed by the researcher. Raw data will be kept in a cupboard, which will be kept under lock and key.

### 3.4. Anonymity

Anonymity refers to the fact that the condition of the researcher is unknown. By using numbers, letters, or codes instead of participants’ names and by not identifying the participants’ conditions or states, including any deformities or clothing based on the information they provide on surveys, the researcher will guarantee participant anonymity.

### 3.5. Privacy

The researcher will protect the privacy of participants by making sure that they are in a close-quarters environment while responding to the questionnaires; the room will be free of people entering and leaving, noise, and disturbances, and personal belongings, such as participant files, will not be used without participants’ permission. Videos or tape recordings will not be used without participants’ knowledge or consent.

### 3.6. Justice

The researcher will ensure that the ethical duty to fairly divide the advantages and costs of the study project is fulfilled. To choose study participants, the researcher will make sure that fair and equal methods are used.

### 3.7. Beneficence

The researcher will make sure that the questionnaires do not include any queries that could compromise the dignity of participants. Through one-on-one meetings in a private setting, the researcher will ensure that any participant who has been traumatized by responding to the research questionnaire receives counseling. Additionally, data collection will be communicated to a psychologist so that the psychologist is prepared to accept individuals who may experience distress.

### 3.8. Autonomy

Participants will be free to choose what they wish to do without worrying about any consequences, thanks to the researcher’s guidance. Participants should make their own well-informed judgments about their participation in the research study, and the researcher will accept that.

### 3.9. Avoidance of Harm

To prevent harm to participants, the researcher will preserve their names and ensure privacy. Counseling will be offered to those who require it, although this study carries minimal risks; regardless, if necessary, participants will be referred to psychologists, with whom the researcher will make arrangements before the study begins.

### 3.10. Plan for Dissemination and Implementation of Results

The results will be published in accredited journals; a copy of the thesis will be available at the university library; and the researcher will return to the PHCs where the data were collected to share the results and recommendations with them.

### 3.11. Declaration of Competing Interest

The researcher declares that they have no interest in competing for financial or personal relationships that could appear to influence the work reported in this paper.

## 4. Discussion

This study examines the knowledge, attitudes, and practices of midwives in the implementation of MHCGs in a selected district, Limpopo province, South Africa. The researcher in this study hopes to learn and discover if there is a relationship between the knowledge of midwives and the implementation of MHCGs. With knowledge, the researcher refers to years of experience and qualifications. The researcher also wanted to identify if the attitudes of midwives somehow affect their practice while implementing MHCGs. As a result, the two theories of Watson Jean’s theory of human care and Everett M. Rodger’s KAP model were incorporated and used as a framework to guide the study. The primary importance of this study stems from the identification of knowledge gaps in midwifery practice, such as identifying if midwives are aware of current maternal health guidelines. We also aim to improve midwives’ attitudes toward the implementation of maternal healthcare guidelines in South Africa.

## Figures and Tables

**Figure 1 nursrep-15-00368-f001:**
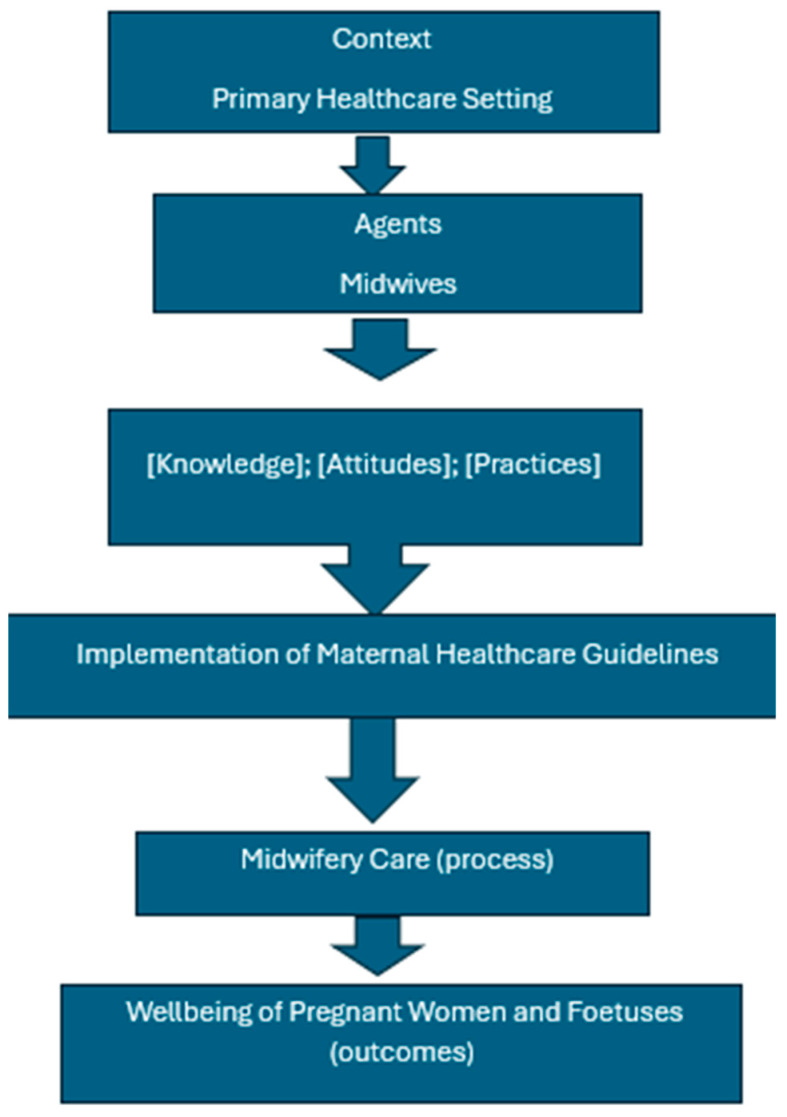
Conceptual Framework Illustrating the Influence of Midwives’ Knowledge, Attitudes, and Practices.

## Data Availability

Data is contained within the article or Appendix A.

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
