# Peer review of "Protocol to Assess the Knowledge, Attitude, and Practices of Midwives in the Implementation of Maternal Healthcare Guidelines in a Selected District, Limpopo Province, South Africa"

_nursrep, 2025, doi:10.3390/nursrep15100368_

Round 1
Reviewer 1 Report
Comments and Suggestions for Authors
Since this study is a mixed method that applies both qualitative and quantitative research, the design of the methodology is very important. However, among them, the design of the methodology for qualitative research is more important than the quantitative research method, but this part seems to be completely missing. This research design seems to explain the basics of research methodology in general, and most of the concepts are defined, but these parts should be deleted and the practical methodology should be written more scientifically so that this study can form basic data to solve current problems well. Therefore, an overall revision is required.
<Introduction>
- The current necessity only states the reality that the maternal mortality rate in the region that they want to continue to study is high. However, this study does not mention at all why they want to investigate the knowledge, attitude, and practice of midwives, and does not mention why they want to conduct qualitative and quantitative research together to clarify these concepts. The researchers should explain why they brought up these three concepts, explain the necessity through supporting literature, and explain why they want to investigate midwives instead of mothers or doctors and what kind of effect they want to achieve.
- framework: I think a conceptual framework would be more appropriate than a theoretical framework. This is because in this study, we will focus on concepts that fit this study rather than on the larger concept of theory. First, it would be good to change the circles to the concepts that this study is trying to identify. It would be good to write midwives instead of human being, wellbeing of pregnant women and the fetuses instead of health, midwifery care instead of nursing, and PHC facility instead of environment. In addition, the current concept map does not explain the contents of the arrows. That is, it is necessary to explain how the arrows go to knowledge, attitude, and practice based on other studies. Currently, there is only a definition of the terms.
- The objectives are too long. It can be summarized in just 2-3 lines, such as simply assessing knowledge, attitudes, and practices and examining their relationships. Please correct the typo in implementation in line 149. And objectives should go in the introduction.
<Methodology>
- There is a need to combine and simplify research methodology and design.
- There is a need to combine study setting and study population.
- 6.1 and 2.6.2 will be combined and simplified.
- The sample size will be based on what statistics and references will be provided for which studies were used to set the degree of error to 0.05.
- Inclusion and exclusion criteria should be explained within the participants. In the inclusion criteria, please state the reason for limiting the participants to those with more than 6 months of experience.
- There is no need to show the questionnaire in the paper. It would be better to erase it.
- You can erase all definitions of face, content, and construct validity and just write a few lines about who you used and how you measured this validity.
- It is said that quantitative and qualitative research will be conducted in parallel, but there is a lack of description of the qualitative research. It should describe what questions will be asked and what kind of training the researchers participating in the qualitative research have received.
- In quantitative research, they say that they will collect fragmentary questions and then use chi-square to identify them, but the results that can come out of this data will be extremely limited. This is because they exist only as nominal or ordinal data. Therefore, finding and utilizing tools that can measure knowledge, attitudes, and practices will bring much more knowledge. Please find those tools and write them down.
Author Response
|
1. Summary |
|
|
Thank you very much for taking the time to review this manuscript. Please find the detailed responses below and the corresponding corrections highlighted in yellow in track changes in the re-submitted files. Comments 1: Introduction The current necessity only states the reality that the maternal mortality rate in the region that they want to continue to study is high. However, this study does not mention at all why they want to investigate the knowledge, attitude, and practice of midwives, and does not mention why they want to conduct qualitative and quantitative research together to clarify these concepts. The researchers should explain why they brought up these three concepts, explain the necessity through supporting literature, and explain why they want to investigate midwives instead of mothers or doctors and what kind of effect they want to achieve. Response 1: Thank you for the suggestion, but this is not a mixed method study. It is a quantitative study. The reason why the researcher chose midwives and not mothers or doctors is because midwives are always the ones who have the first encounter with the pregnant mothers, and in most cases, they care for pregnant women from the time of their first ANC visit up until they deliver without the help of a doctor. Pregnant women will not be considered in this case because the maternal healthcare was designed for healthcare professionals, not patients. Comments 2: Framework: I think a conceptual framework would be more appropriate than a theoretical framework. This is because in this study, we will focus on concepts that fit this study rather than on the larger concept of theory. First, it would be good to change the circles to the concepts that this study is trying to identify. It would be good to write midwives instead of human being, wellbeing of pregnant women and the fetuses instead of health, midwifery care instead of nursing, and PHC facility instead of environment. In addition, the current concept map does not explain the contents of the arrows. That is, it is necessary to explain how the arrows go to knowledge, attitude, and practice based on other studies. Currently, there is only a definition of the terms. Response 2: We agree with this comment. We wrote the conceptual framework as suggested. Comments 3: The objectives are too long. It can be summarized in just 2-3 lines, such as simply assessing knowledge, attitudes, and practices and examining their relationships. Please correct the typo in implementation in line 149. And objectives should go in the introduction. Response 3: thank you for mentioning the typing error, we corrected it. Objectives are summarized. Comments 4: Methodology There is a need to combine and simplify research methodology and design. Response 4: Research methodology and design combined. Comments 5: There is a need to combine study setting and study population. Response 5: Study setting and study population combined. Comments 6: 2.6.1 and 2.6.2 will be combined and simplified. Response 6: thank you for the suggestion. We combined them. Comment 7: The sample size will be based on what statistics and references will be provided for which studies were used to set the degree of error to 0.05. Response 7: The sample size was going to be determined using Taro Yamane’s theory. The 0.05 degree of error if from the above mentioned formular which will be utilized in this study, but the second reviewer suggested we use GPower software to determine sample size, so we wont be using the formular. Comments 8: Inclusion and exclusion criteria should be explained within the participants. In the inclusion criteria, please state the reason for limiting the participants to those with more than 6 months of experience. Response 8: The reason for limiting the participants to those that have more that 6 months of experience is because they have enough exposure to maternal healthcare guidelines and they offer maternal healthcare services to pregnant women. We highlighted this in the manuscript. Comment 9: There is no need to show the questionnaire in the paper. It would be better to erase it. Response 9: Thank you for your suggestion, but the comments from the second reviewer helped to improve the questionnaire and we think its better we keep it and not remove it. Comments 10: You can erase all definitions of face, content, and construct validity and just write a few lines about who you used and how you measured this validity. Response 10: Thanks for the suggestion, all definitions erased. Comments 11: It is said that quantitative and qualitative research will be conducted in parallel, but there is a lack of description of the qualitative research. It should describe what questions will be asked and what kind of training the researchers participating in the qualitative research have received. Response 11: Thank you for the suggestion, but this is a quantitative method only study. Comments 12: In quantitative research, they say that they will collect fragmentary questions and then use chi-square to identify them, but the results that can come out of this data will be extremely limited. This is because they exist only as nominal or ordinal data. Therefore, finding and utilizing tools that can measure knowledge, attitudes, and practices will bring much more knowledge. Please find those tools and write them down. Response 12: Noted, we made use of Likert scale, yes or no questions, and semi-structured questions
|
|

Reviewer 2 Report
Comments and Suggestions for Authors
Overall, this was an interesting protocol. Most sections were well described, particularly the introduction, however, the methodology had a number of shortcomings. The design of the study has not been effectively communicated and there appears to be some misunderstanding or hesitancy with respect to the statistical aspects of the proposed study. For example, what are the independent and dependent variables? The authors mention chi-square which examines a significant relationship between two categorical variables, yet do not explicitly state what these categorical variables are. Moreover, it is recommended to compute sample size using standard power analysis tools. I have provided a link to a free tool below.
There are also a number of instances where sections have generic information about study design, with no explicit relevance to the present study. Therefore, it is advised that this is modified throughout. There is no need to outline what a quantitative and qualitative approach is - here you can simply state you will use a quantitative cross-sectional approach.
With respect to the questionnaire that has been developed I have a number of questions – will you use cronbach’s alpha to assess reliability? Most importantly, how is each section of the questionnaire scored? Presumably you will generate a knowledge score and attitude score? I must note that the number of questions is rather small. It is also beneficial to explain how these questions were generated.
Lastly, it would be beneficial to explicitly state in your discussion why the study is necessary and the potential impact it could have on the practices of Midwives in South Africa and perhaps the ability to inform future educational interventions?
Please see below for an overview of comments per section:
Title:
Well written – perhaps including ‘A protocol’ into this would be beneficial.
Abstract:
Well written – the focus of the study is clearly outlined, including design, location and sampling method. Is it necessary to state results will be available after the collection of data as this is self-evident?
Introduction:
Please remove the capitalised ‘p’ in ‘poor’ line 47.
Please could the authors elaborate on ‘previously learned knowledge’ (line 55).
Please provide some examples of what the evidence-based guidelines for the management of maternal diseases includes (line 55).
It would be beneficial to be more explicit when outlining the reasons maternal deaths occur – for example, what is meant by ‘poor management skills’ specifically?
Materials and methods
The objectives are sound and map on to the focus previously outlined.
Research methodology
It is unnecessary here to outline qualitative research. Here you can simply state that you will employ a quantitative cross-sectional design.
Research design
The content included here could be improved. Outlining what the predictors/outcome variables are, is needed. Moreover, information about the questionnaires to be used, number of items etc. would be best placed here.
Sampling
There are three sections for sampling which is unnecessary. Please merge these into one single section that clearly outlines the sampling approach undertaken. At present, this is very confusing and oddly structured.
Sample size
It is advised that the authors use G*Power software to conduct their power analysis based on their proposed design. This can be downloaded for free, here: https://www.psychologie.hhu.de/arbeitsgruppen/allgemeine-psychologie-und-arbeitspsychologie/gpower
Data collection
Section A
Would it be worth including gender in case some midwives are male?
Section B
Some questions are too broad. For example, ‘I encounter problems when reading maternal healthcare guidelines’. It would be optimal for the authors to then include a follow-up question where if participants responded ‘strongly agree’ or ‘agree’ a free-response text box is included so participants can outline what problems they encounter.
Section C
Item 2 – this should be made more specific as opposed to ‘not enough for all of us’. Who is ‘us’ here?
Data analysis
Please outline scoring instructions for your questionnaires (specifically sections B and C). How will knowledge be calculated and what do higher scores indicate? The same question can be posited for attitudes.
Will the authors conduct cronbach’s alpha to assess the reliability of these questionnaires?
Discussion
The discussion could be improved to re-state the importance and potential contribution of the study and the impact it could have on the practices of midwives in South Africa.
Author Response
|
Response to Reviewer 2 Comments
|
||
|
1. Summary |
|
|
|
Thank you very much for taking the time to review this manuscript. Please find the detailed responses below and the corresponding corrections highlighted in yellow in track changes in the re-submitted files.
|
||
|
Comments 1: Title: Well written – perhaps including ‘A protocol’ into this would be beneficial. |
||
|
Response 1: Done. We agree with this comment.
|
||
|
Comments 2: Abstract Well written – the focus of the study is clearly outlined, including design, location and sampling method. Is it necessary to state results will be available after the collection of data as this is self-evident? |
||
|
Response 2: we removed the sentence, kindly check the abstract.
Comments 3: Introduction: Please remove the capitalised ‘p’ in ‘poor’ line 47. Please could the authors elaborate on ‘previously learned knowledge’ (line 55). Please provide some examples of what the evidence-based guidelines for the management of maternal diseases includes (line 55). It would be beneficial to be more explicit when outlining the reasons maternal deaths occur – for example, what is meant by ‘poor management skills’ specifically?
Response 3: capitalized P removed. Kindly check page 2. - By saying previously learned knowledge, the authors meant to say that midwives are managing women with the knowledge they got during their basic midwifery training which could be years ago, and things are evolving in the maternity healthcare guidelines. - Examples of evidence-based guidelines for management of maternal conditions included. - We mentioned some examples of poor management skills by healthcare professionals.
Comments 4: Research methodology It is unnecessary here to outline qualitative research. Here you can simply state that you will employ a quantitative cross-sectional design. Response 4: noted and resolved. See page 4.
Comments 5: Research design The content included here could be improved. Outlining what the predictors/outcome variables are, is needed. Moreover, information about the questionnaires to be used, number of items etc. would be best placed here. Response 5: Noted, but the information about the questionnaires is indicated under the data collection section.
Comments 6: Sampling There are three sections for sampling which is unnecessary. Please merge these into one single section that clearly outlines the sampling approach undertaken. At present, this is very confusing and oddly structured. Response 6: Thanking you for suggesting that. All sections merged into one section and thoroughly explained.
Comments 7: Sample size It is advised that the authors use G*Power software to conduct their power analysis based on their proposed design. This can be downloaded for free, here: https://www.psychologie.hhu.de/arbeitsgruppen/allgemeine-psychologie-und-arbeitspsychologie/gpower. Response 7: Thank you for the advice, we will make use of G Power software to determine the sample size
Comments 8: Data collection Section A Would it be worth including gender in case some midwives are male? Response 8: gender included.
Comments 9: Section B Some questions are too broad. For example, ‘I encounter problems when reading maternal healthcare guidelines. It would be optimal for the authors to then include a follow-up question where if participants responded ‘strongly agree’ or ‘agree’ a free-response text box is included so participants can outline what problems they encounter. Response 9: A follow up question included.
Comments 10: Section C Item 2 – this should be made more specific as opposed to ‘not enough for all of us’. Who is ‘us’ here? Response 10: Thank you for pointing this out, “all of us” referred to midwives. We corrected it.
Comments 11: Data analysis Please outline scoring instructions for your questionnaires (specifically sections B and C). How will knowledge be calculated and what do higher scores indicate? The same question can be posited for attitudes. Will the authors conduct Cronbach’s alpha to assess the reliability of these questionnaires? Response 11: The scoring instructions outlined, and yes, the author will conduct Cronbach’s alpha to assess the reliability of questionnaires.
Comments 12: Discussion The discussion could be improved to re-state the importance and potential contribution of the study and the impact it could have on the practices of midwives in South Africa. Response 12: We agree with this comment. Therefore, we added more information on the discussion. |
||

Round 2
Reviewer 1 Report
Comments and Suggestions for Authors
First, you should submit the manuscript without showing the revised parts. You can just highlight the changed parts, but the readability was very poor because the lines were visible due to the revisions here and there, which made the second review quite difficult. I regret this, so I ask that researchers turn off the change tracking function in the future.
Since you said that you will conduct the research only as a quantitative study, the last comment became much more important. However, it seems that you did not make any changes to this. The data collected through this questionnaire will only provide very shallow information. As a reviewer who diligently read and revised this research protocol, I have many regrets. I am worried whether this protocol will be helpful to other researchers and will be cited.
Author Response
comments 1:
First, you should submit the manuscript without showing the revised parts. You can just highlight the changed parts, but the readability was very poor because the lines were visible due to the revisions here and there, which made the second review quite difficult. I regret this, so I ask that researchers turn off the change tracking function in the future.
Since you said that you will conduct the research only as a quantitative study, the last comment became much more important. However, it seems that you did not make any changes to this. The data collected through this questionnaire will only provide very shallow information. As a reviewer who diligently read and revised this research protocol, I have many regrets. I am worried whether this protocol will be helpful to other researchers and will be cited.
Response 1: thank you for pointing this out, we agree with you. in the last comment you mentioned that we should find the tools that will measure knowledge, attitudes and practices and we did mention them in the data analysis. we have turned off tracking and highlighted the changed parts.

Reviewer 2 Report
Comments and Suggestions for Authors
This is a much improved protocol - thank you for comprehensively addressing my comments.
Author Response
Thank for your feedback.
